# Interactions between Cultivated *Gracilariopsis lemaneiformis* and Floating *Sargassum horneri* under Controlled Laboratory Conditions

**Hanmo Song, Yan Liu \*, Jingyu Li, Qingli Gong and Xu Gao \***

Key Laboratory of Mariculture (Ministry of Education), Fisheries College, Ocean University of China, Qingdao 266003, China

\* Correspondence: qd_liuyan@ouc.edu.cn (Y.L.); gaoxu@ouc.edu.cn (X.G.);
Tel.: +86-532-8203-2377 (Y.L. & X.G.)

**Abstract:** The golden tide dominated by *Sargassum* has become a frequently-occurring marine ecological event that may constitute major biotic threats to seaweed aquaculture. In this study, the interaction between cultivated *Gracilariopsis lemaneiformis* (GL) and floating *Sargassum horneri* (SH) was investigated by physiological and biochemical measurements under mono-culture and co-culture with different biomass density ratios of 2:1 (2GL:1SH), 1:1 (1GL:1SH), and 1:2 (1GL:2SH). The relative growth rate, net photosynthetic rate, and $NO_3$-N uptake rate of *G. lemaneiformis* were significantly greater at the biomass density ratio of 2:1 than at mono-culture. However, these physiological parameters and biochemical composition contents (chlorophyll *a* and soluble protein) of *G. lemaneiformis* decreased significantly with increasing biomass of *S. horneri*. Meanwhile, these physiological and biochemical parameters of *S. horneri* were greater in all co-culture models than at mono-culture. They decreased significantly with decreasing biomass of *G. lemaneiformis*. These results indicate that the occurrence of floating *S. horneri* with low biomass can stimulate the growth of *G. lemaneiformis*, whereas its outbreak may significantly reduce the production and quality of *G. lemaneiformis*. *G. lemaneiformis* cultivation may be beneficial to the increased biomass of floating *S. horneri*.

**Keywords:** biochemical composition; golden tide; growth; $NO_3$-N uptake; photosynthesis; seaweed cultivation

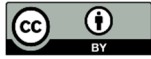

## 1. Introduction

In recent years, multiple large-scale outbreaks of golden tide caused by *Sargassum* have occurred along the coastal waters of the Pacific and Atlantic Ocean [1–4]. This phenomenon may be closely correlated with marine ecological deterioration, including seawater eutrophication and warming [5,6]. Along the northwest Pacific coast, *Sargassum horneri* is a predominant golden-tide-forming seaweed that originally forms extensive underwater forests [7,8]. Sessile populations of *S. horneri* have continuously decreased in this region [9], whereas a prominent drifting biomass has been frequently detected in the coastal regions of China in the last decade [3,10]. Golden tides have caused considerable disruption to coastal ecosystems, including the death of aquatic organisms by hypoxia, resource competition with native species, and even shifts in biological community structures [11,12]. Furthermore, it has brought considerable damage to local economic industries such as tourism, traditional fisheries, and mariculture [13–15].

Positive and negative interactions among plants are widespread in both terrestrial and aquatic ecosystems, and have complicated and changeable effects on population dynamics, community structure, and biodiversity patterns [16–18]. In coastal regions, a com-

mon interspecific interaction pattern is for marine plants with overlapping niches to compete for diverse limited environmental resources [19–21]. Epiphytic *Ulva lactuca* has been proven to severely interfere with mariculture of *Neoporphyra haitanensis*, as they compete for carbon and nitrogen resources [22]. Additionally, the growth of *Neopyropia yezoensis* was apparently restrained by *Ulva prolifera* proliferation because of competition for space, nutrients, and light for spore attachments [23,24]. Allelopathy is another pattern of interspecific interaction among marine algae, which can stimulate or inhibit each other through the release of biochemical metabolites [25–27]. Yuan et al. [28] showed that aqueous extract from *Sargassum fusiforme* promoted the growth of *Karenia mikimotoi* at 0.2 g L$^{-1}$, while its growth was inhibited at 1.6 g L$^{-1}$. Similarly, Patil et al. [29] documented that the growth of *Skeletonema costatum* was not significantly decreased by culture filtrates of *Pyropia haitanensis* at 0.625–10 g FW L$^{-1}$, while it was greatly inhibited at 15 and 20 g FW L$^{-1}$. These results imply that interaction through allelopathy is closely associated with the biomass density ratio among species. However, thus far there has been little information produced regarding the influence of the biomass density ratio on the interaction between marine macroalgae.

*Gracilariopsis lemaneiformis* is a common intertidal red seaweed that is naturally distributed across temperate regions worldwide [30–32]. In China, *G. lemaneiformis* is extensively cultivated due to its widespread applications in agar production, the food industry, and abalone aquaculture [33–35]. It is the seaweed species with the third highest production rate in terms of mass, with 368,967 tons of dry weight produced in 2020 [36]. Additionally, the cultivation of this species is beneficial to coastal ecosystems in many ways, including alleviation of harmful microalgal blooms, increasing the dissolved oxygen in the seawater, and maintaining coastal ecological balance [37]. In recent years, the deterioration of the marine environment caused by climate change and human activities has posed a severe threat to the mariculture production of seaweeds [38–40]. Due to the economic and ecological importance of *G. lemaneiformis*, a considerable number of studies about the impacts of diverse abiotic factors on its physiological and biochemical performance have been conducted [41–44]. Nevertheless, the effects of biotic stresses such as macroalgal blooms on *G. lemaneiformis* have rarely been reported.

In the current study, we performed a short-term culture experiment to examine the interactive influences between cultivated *G. lemaneiformis* and floating *S. horneri*. Changes in growth, photosynthesis, NO$_3$-N uptake, and biochemical compositions were estimated. Our results provide significant information to improve the mariculture management of this valuable species and to address marine ecological hazards.

## 2. Materials and Methods

### 2.1. Sampling and Maintenance

*G. lemaneiformis* thalli were sampled from farmed populations on Lidao Island (36°26′ N, 122°56′ E), China, in June 2020. Meanwhile, *S. horneri* thalli were sampled from floating populations in the cultivation area of *G. lemaneiformis*. Healthy samples were chosen and completely rinsed using sterilized seawater to remove epiphytes and detritus. Algal fragments of *G. lemaneiformis* (6 cm in length) and *S. horneri* (3 cm in length) were respectively excised from the apical position of branches for the experiments. Then, they were incubated in tanks containing 6 L of 25% PESI medium [45]. These fragments were kept for three days at 20 °C, 90 μmol photons m$^{-2}$ s$^{-1}$, and a 12:12 L:D cycle in order to minimize the impacts of excision.

### 2.2. Culture Experiment

A twelve-day culture experiment was performed under mono-culture conditions of *G. lemaneiformis* and *S. horneri* and co-culture at different biomass density ratios (BDRs). For the mono-culture, the initial biomass density was set at 6 g for both species. For the co-culture, the initial BDR was set at 2:1 (4 g *G. lemaneiformis* cultured with 2 g *S. horneri*),

1:1 (3 g *G. lemaneiformis* cultured with 3 g *S. horneri*), and 1:2 (2 g *G. lemaneiformis* cultured with 4 g *S. horneri*). A total of five treatments were set up, and each treatment was conducted in three replicates. In this experiment, 20 °C, 90 μmol photons $m^{-2} s^{-1}$, and a 12:12 L:D cycle were maintained. This experiment used fifteen tanks, with each tank containing 6 L of sterilized seawater enriched with 25% PESI medium. Gentle aeration was conducted during this experiment. The medium was changed every three days. The fresh weights (FWs) of all fragments before and after incubation were determined after each fragment was blotted dry. The relative growth rate (RGR; % $day^{-1}$) was calculated by Equation (1):

$$RGR = 100 \times (lnW_t - lnW_0) / t \tag{1}$$

where $W_0$ is the initial FW, $W_t$ is the final FW, and t is the culture time in days.

### 2.3. Measurement of Photosynthesis

Following the culture experiment, we determined the net photosynthetic rates of *G. lemaneiformis* and *S. horneri* using a manual FireStingO$_2$II oxygen meter (Pyro Science GmbH, Aachen, Germany). For each species, 0.33 g (FW) of samples from each treatment were moved to the oxygen electrode cuvette containing 330 mL of 25% PESI medium. Next, the medium was continuously stirred to ensure homogenous oxygen diffusion. The culture conditions were the same as for the experiment above. The oxygen increase in the seawater was regarded as the net photosynthetic rate after an increase in the light density. Before measurements, these samples were placed into the cuvette for 5 min acclimation. The oxygen concentration in the medium was documented every 1 min for 15 min.

### 2.4. Measurement of NO$_3$-N Uptake Rate

Following the culture experiment, 0.2 g (FW) of *G. lemaneiformis* and/or *S. horneri* were randomly selected from each tank for NO$_3$-N uptake measurements. They were transferred into conical flasks containing 200 mL of sterilized seawater enriched with 25% PESI medium and gently shaken for 2 h using a horizontal oscillator. The light and temperature conditions were the same as those described for the culture experiment. For each treatment, the media before and after the culture experiment were separately collected and the NO$_3$-N concentrations were determined by the cadmium column reduction method and molybdenum blue method with ultraviolet absorption spectrophotometer, respectively [46,47]. The NO$_3$-N uptake rate was calculated by Equation (2):

$$U_N = (C_0 - C_t) \times V / (T \times W) \tag{2}$$

where $U_N$ is the NO$_3$-N uptake rate, $C_0$ is the initial concentration of NO$_3$-N (mg $L^{-1}$), $C_t$ is the final concentration of NO$_3$-N (mg $L^{-1}$) following the experiment, V is the medium volume (mL), T is the experiment period, and W is the FW of samples (g).

### 2.5. Measurement of Chlorophyll a (Chl a)

0.1 g (FW) of *G. lemaneiformis* was used to extract the Chl *a* in each replicate of all the experimental treatments. These samples were ground using liquid nitrogen, then 3 mL of phosphate buffer solution (0.1 M, pH = 6.8) was added. After transferring the samples into tubes, they were centrifuged for 30 min at 4000 rpm at 4 °C. Subsequently, 8 mL of dimethylformamide (DMF) was added to the precipitates obtained following centrifugation and maintained at 4 °C for 1 d. Extracts were then centrifuged for 10 min at 6000 rpm at 4 °C. The supernatant was collected and the absorption was determined at 750, 664, 647, and 625 nm. The Chl *a* content (mg $g^{-1}$) was calculated by Equation (3):

$$Chl\ a = [12.65 \times (A_{664} - A_{750}) - 2.99 \times (A_{647} - A_{750}) - 0.04 \times (A_{625} - A_{750})] \times V_e / (I \times W \times 1000) \tag{3}$$

where $A_{750-625}$ is the absorption of extracts at 750, 664, 647, and 625 nm, $V_e$ is the DMF volume (mL), I is the optical path in the cuvette (cm), and W is the FW of the samples (g).

0.25 g (FW) of *S. horneri* was used to extract the Chl *a* in each replicate of all the experimental treatments. These samples were placed into 2 mL of dimethyl sulfoxide for 5 min, and the supernatant absorption was determined at 665, 631, 582, and 480 nm. Subsequently, the same samples were placed into 3 mL of acetone for 2 h. The supernatant was moved to a 10 mL tube, then 1 mL of methanol and 1 mL of distilled water were added. The supernatant absorption was determined at 664, 631, 581, and 470 nm. The Chl *a* content (mg g$^{-1}$) was calculated by Equation (4):

$$\text{Chl } a = [(A_{665} / 72.8) \times V_1 + (A_{664} / 73.6) \times V_2] / (I \times W) \tag{4}$$

where $A_{665}$ and $A_{664}$ are the absorption rates of the extracts at 665 and 664 nm, respectively, $V_1$ is the volume of the extract at the first extraction process (mL), $V_2$ is the volume of extract at the second extraction process (mL), I is the optical path in the cuvette (cm), and W is the FW of the samples (g).

### 2.6. Measurements of Soluble Protein and Carbohydrate

For all the experimental treatments, 0.1 g (FW) of samples from each replicate were homogenized with a pestle and mortar and 0.9 mL of phosphate buffer solution (0.1 M, pH = 7.4). The extract was centrifuged for 30 min at 12,000 rpm at 4 °C. The supernatant absorption at 595 nm was recorded using an ultraviolet spectrophotometer. The soluble proteins of samples were evaluated using Coomassie Brilliant Blue G-250 dye (Nanjing Jiancheng Bioengineering Institute, Nanjing, China) and bovine albumin [48].

An amount (0.1 g FW) of the samples from each replicate were ground in 2 mL of distilled water and diluted to 10 mL after adding 2 mL of $MgCO_3$ suspension liquid. The mixture was centrifuged for 5 min at 4000 rpm at 4 °C. Then, 1 mL of supernatant was moved to a tube and diluted to 2 mL. After that, 8 mL of anthrone reagent was added. The mixture was bathed in boiled water for 10 min and cooled to room temperature. The absorption at 620 nm was recorded and the mixture was standardized. The soluble carbohydrate content was measured according to [49].

### 2.7. Statistical Analysis

For both species, one-way ANOVA was applied to analyze the RGR, net photosynthetic rate, $NO_3$-N uptake rate, and the contents of biochemical compositions among different culture models (mono- or co-cultures with three BDRs). Prior to these analyses, the variances of the data were subjected to homogeneity tests. Duncan's multiple range test was used if significant differences were detected ($p < 0.05$). Statistical analyses were carried out using SPSS 26.0 software.

## 3. Results

### 3.1. Growth

The RGRs of *G. lemaneiformis* were significantly affected by culture model (Figure 1A and Table 1). The RGR at BDR of 2:1 was significantly greater than that at mono-culture. However, the RGR decreased significantly with increasing biomass of *S. horneri*, from BDR of 2:1 to 1:2. The RGRs of *S. horneri* differed significantly among culture models (Figure 1B and Table 1). The RGRs of all co-culture treatments were greater than that at mono-culture, and significantly higher values were found at BDRs of 2:1 and 1:1. Additionally, the RGR at BDR of 2:1 was significantly greater than that at BDR of 1:1.

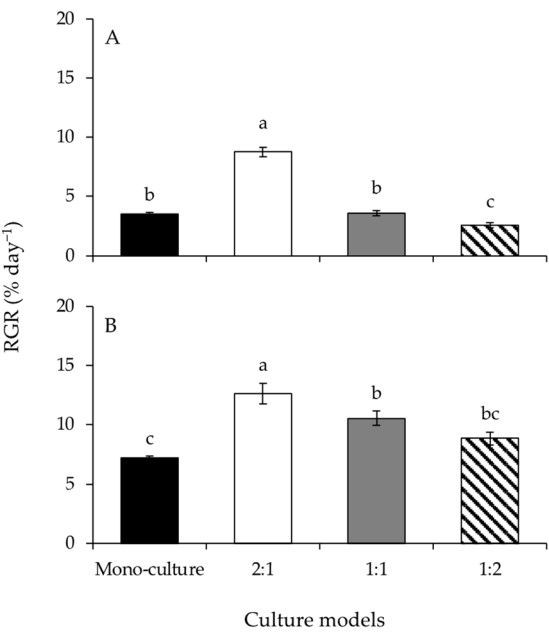

**Figure 1.** The RGRs of *Gracilariopsis lemaneiformis* (**A**) and *Sargassum horneri* (**B**) cultured for twelve days at mono-culture and co-culture with biomass density ratios of 2:1, 1:1, and 1:2. Different lowercase letters represent significant differences ($p < 0.05$) among culture models.

**Table 1.** One-way ANOVA testing of the effects of biomass density ratio on RGR, net photosynthetic rate, $NO_3$-N uptake rate, Chl *a*, soluble protein, and carbohydrate of *Gracilariopsis lemaneiformis* and *Sargassum horneri*.

| Factors | *Gracilariopsis lemaneiformis* | | | *Sargassum horneri* | | |
|---|---|---|---|---|---|---|
| | **df** | **F** | ***p*** | **df** | **F** | ***p*** |
| RGR | 3 | 112.721 | <0.001 | 3 | 16.595 | <0.001 |
| Net photosynthetic rate | 3 | 29.975 | <0.001 | 3 | 6.836 | <0.05 |
| $NO_3$-N uptake rate | 3 | 52.763 | <0.001 | 3 | 32.004 | <0.001 |
| Chl *a* | 3 | 4.510 | <0.05 | 3 | 4.157 | <0.05 |
| Soluble protein | 3 | 6.398 | <0.05 | 3 | 11.253 | <0.01 |
| Soluble carbohydrate | 3 | 39.980 | <0.001 | 3 | 33.784 | <0.001 |

*3.2. Photosynthesis*

The net photosynthetic rates of *G. lemaneiformis* differed significantly among culture models (Figure 2A and Table 1). The net photosynthetic rate at BDR of 2:1 was significantly higher than those of the other culture models. Additionally, the net photosynthetic rate at BDR of 1:2 was significantly lower than that at mono-culture. The net photosynthetic rates of *S. horneri* were significantly affected by culture model (Figure 2B and Table 1). The net photosynthetic rate at BDR of 2:1 was significantly higher than that at mono-culture. However, the net photosynthetic rates decreased significantly with increasing biomass of *S. horneri*, from BDR of 2:1 to 1:2.

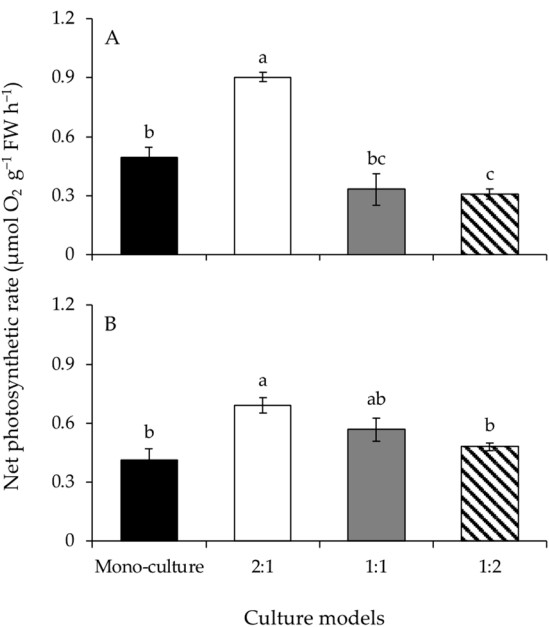

**Figure 2.** The net photosynthetic rates of *Gracilariopsis lemaneiformis* (**A**) and *Sargassum horneri* (**B**) cultured for twelve days at mono-culture and co-culture with biomass density ratios of 2:1, 1:1, and 1:2. Different lowercase letters represent significant differences (*p* < 0.05) among culture models.

### 3.3. NO₃-N Uptake Rate

A significant effect of the culture model on the $NO_3$-N uptake rates of *G. lemaneiformis* was detected (Figure 3A and Table 1). The $NO_3$-N uptake rate at BDR of 2:1 was significantly greater than those of the other culture models. The $NO_3$-N uptake rates of *S. horneri* were significantly different among culture models (Figure 3B and Table 1). The $NO_3$-N uptake rates of the co-culture treatments were significantly greater than that at mono-culture. The $NO_3$-N uptake rates decreased significantly with increasing biomass of *S. horneri*, from BDR of 2:1 to 1:2.

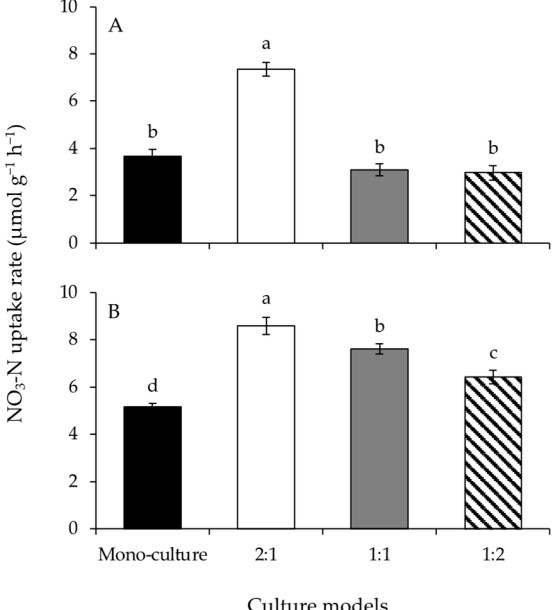

**Figure 3.** The $NO_3$-N uptake rates of *Gracilariopsis lemaneiformis* (**A**) and *Sargassum horneri* (**B**) cultured for twelve days at mono-culture and co-culture with biomass density ratios of 2:1, 1:1, and 1:2. Different lowercase letters represent significant differences (*p* < 0.05) among culture models.

### 3.4. Chl a

The Chl *a* contents of *G. lemaneiformis* differed significantly among culture models (Figure 4A and Table 1). The Chl *a* content at BDR of 1:2 was significantly lower than at mono-culture and BDR of 2:1. Similarly, the Chl *a* contents of *S. horneri* differed significantly among culture models (Figure 4B and Table 1). The Chl *a* content at BDR of 2:1 was significantly greater than at mono-culture and at BDR of 1:2.

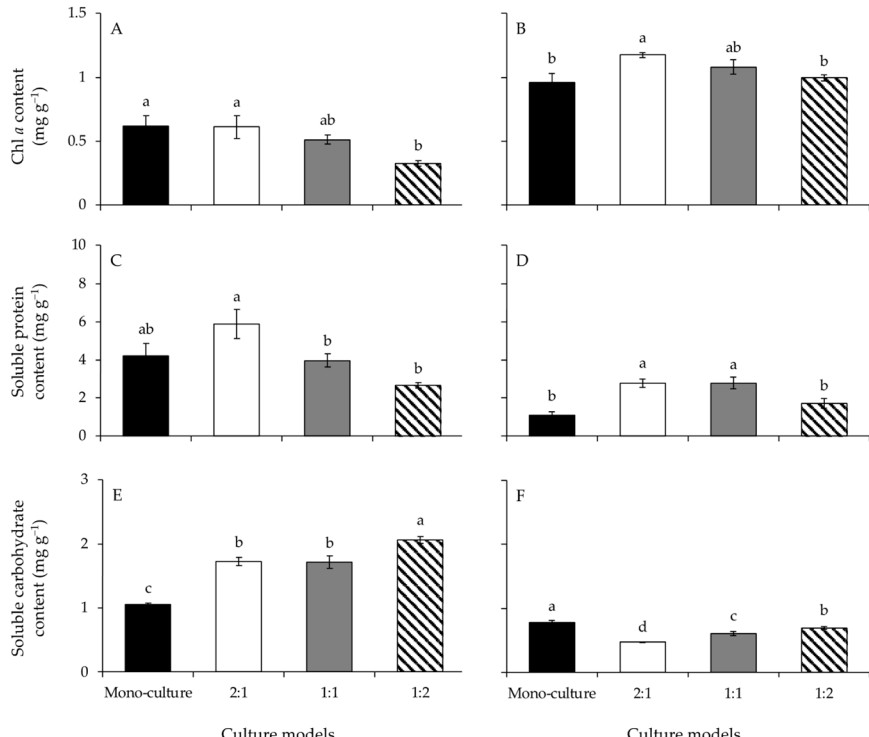

**Figure 4.** The contents of Chl *a*, soluble protein, and soluble carbohydrate of *Gracilariopsis lemaneiformis* (**A,C,E**) and *Sargassum horneri* (**B,D,F**) cultured for twelve days at mono-culture and co-culture with biomass density ratios of 2:1, 1:1, and 1:2. Different lowercase letters represent significant differences (*p* < 0.05) among culture models.

### 3.5. Soluble Protein and Carbohydrate

The one-way ANOVA results showed a significant difference among culture models on the soluble protein contents of *G. lemaneiformis* (Figure 4C and Table 1). The soluble protein content at BDR of 2:1 was significantly greater than at BDRs of 1:1 and 1:2. The soluble protein contents of *S. horneri* differed significantly among culture models (Figure 4D and Table 1). The soluble protein contents at BDRs of 2:1 and 1:1 were significantly greater than at mono-culture and BDR of 1:2.

A significant effect of the culture model on the soluble carbohydrate contents of *G. lemaneiformis* was detected (Figure 4E and Table 1). Compared to the mono-culture, the soluble carbohydrate content had significantly higher values at co-culture with different BDRs. Moreover, the soluble carbohydrate content at BDR of 1:2 was significantly greater than at BDRs of 2:1 and 1:1. Similarly, the soluble carbohydrate contents of *S. horneri* varied significantly among culture models (Figure 4F and Table 1). The soluble carbohydrate content was significantly higher at mono-culture than co-culture with different BDRs. The soluble carbohydrate contents increased significantly with increasing biomass of *S. horneri*, from BDR of 2:1 to 1:2.

## 4. Discussion

Diverse positive and negative interspecific interactions exist among plants in natural communities [50]. The principal mechanisms of these interactions, such as resource competition or allelopathy, are often species-specific and environment-dependent [51,52]. In this study, the RGR of *G. lemaneiformis* was significantly enhanced under co-culture with BDR of 2:1 compared to under mono-culture, indicating that the presence of *S. horneri* with a relatively lower biomass had a stimulating impact on *G. lemaneiformis*. A similar finding was documented for *G. lemaneiformis* and epiphytic *U. prolifera*, showing that the RGR of *U. prolifera* was greatly increased under co-culture with BDR of 1:1 [53]. This 'hormesis' effect may have resulted from the division and expansion of cells through the enzymatic or non-enzymatic processes following low-dose phytotoxin stimulation, thereby further increasing the growth of thalli [54,55]. Additionally, this may be due to an overcompensation response following the initial damage caused by low concentrations of phytotoxins [56,57]. Therefore, we speculated that the secretion of allelochemicals played a more critical role in interspecific interactions than resource competition when *G. lemaneiformis* suffered from a low biomass value of *S. horneri*. Moreover, allelopathy presents complex mechanisms that can influence a variety of physiological and biochemical processes [58]. In this study, we observed a significant increase in the net photosynthetic rate, $NO_3$-N uptake rate, and soluble protein content of *G. lemaneiformis* under co-culture with BDR of 2:1. Similarly, Pan et al. [53] showed that, compared to mono-culture, the Chl *a* content and photosynthetic rate of *G. lemaneiformis* were significantly enhanced under co-culture with 0.5 g $L^{-1}$ of *U. prolifera*. Further experiments are required to clarify the correlative mechanisms of these findings.

Compared to the stimulating effect of *S. horneri* on *G. lemaneiformis* with a relatively lower biomass, the RGR at BDR of 1:2 was significantly lower than at mono-culture. Similarly, Xie et al. [59] demonstrated that the initial population density rate greatly influences the interaction of *Heterosigma akashiwo* (H) and *Prorocentrum donghaiense* (P). The growth of *H. akashiwo* was restrained by *P. donghaiense* at the inoculation proportion of 1H:4P, whereas *H. akashiwo* possessed a higher growth rate at the inoculation proportion of 4H:1P. Both nutrient competition and allelopathy have been demonstrated to play a critical role in interspecific interaction between these two microalgal species. In addition, the net photosynthetic rate, Chl *a* content, and soluble protein content of *G. lemaneiformis* significantly decrease at BDR of 1:2. A similar result was presented in [24], where the growth, photosynthetic activity, and biochemical metabolism of *N. yezoensis* were significantly inhibited by epiphytic *U. prolifera*, mainly as a result of nutrition and light competition. On the other hand, high concentrations of allelochemicals may lead to cell expansion and rupture, while cells can maintain osmotic pressure and intracellular homeostasis by accumulating soluble carbohydrates [60–62]. In the current study, the soluble carbohydrate content of *G. lemaneiformis* presented a significant increase at BDR of 1:2, indicating that *S. horneri* may have exerted a strong allelopathic effect on *G. lemaneiformis* under this condition. Therefore, the inhibitory effect of *S. horneri* with a relatively higher biomass on *G. lemaneiformis* may be associated with a combination of resource competition and allelopathy.

Compared to the mono-culture, the RGR, net photosynthetic rate, $NO_3$-N uptake rate, Chl *a* content, and soluble protein content of *S. horneri* were enhanced under co-culture conditions with *G. lemaneiformis* at different BDRs. Macroalgal species with a stronger growth capacity usually have advantages in interspecific competition scenarios [51,63]. Our results showed that *S. horneri* (7.23%) has a better growth rate than *G. lemaneiformis* (3.53%) under mono-culture conditions. The outstanding growth performance of *S. horneri* may be correlated with its adaptability to frequent marine environment changes during the long-distance floating processes [64]. Additionally, *U. lactuca* showed greater growth and maximal photosynthetic ability than *G. lemaneiformis* under competitive conditions, benefiting from its larger frond surface area and higher nutrient uptake rate [65]. All these results indicate that *S. horneri* has a dominant role in its competition with *G. lemaneiformis* because of its physiological and morphological advantages. In addition, we observed that

all physiological and biochemical parameters of *S. horneri* increased significantly with increasing biomass of *G. lemaneiformis* under co-culture conditions. Wang et al. [66] reported that co-culture with a small quantity of *G. lemaneiformis* thalli was significantly beneficial to the growth of the red tide species *H. akashiwo*. Similarly, *G. lemaneiformis* can stimulate the growth of *S. costatum* by secreting various types of allelochemicals [67]. Therefore, we suggest that certain allelochemicals produced by *G. lemaneiformis* may partially contribute to the increased growth rate of *S. horneri*.

The golden tide caused by *Sargassum* species has become a major ecological threat to the coastal environment and aquaculture production. According to the present data, although the occurrence of small quantities of *S. horneri* can stimulate the growth of *G. lemaneiformis,* its outbreak has a greatly harmful impact on the growth and biochemical metabolism of *G. lemaneiformis*, resulting in reduced yield and quality. Inversely, the growth of *S. horneri* is enhanced by the occurrence of *G. lemaneiformis*, regardless of BDR. This observation suggests that *G. lemaneiformis* cultivation has no inhibitory effect on golden tide, rather favoring its further expansion. Hence, *S. horneri* should be removed or intercepted in time to minimize economic and ecological losses. Due to the limited availability of data in this study, more studies involving related physiological and molecular mechanisms are required to evaluate the coping strategies of *G. lemaneiformis* against golden tide outbreaks.

**Author Contributions:** Investigation, data curation, visualization, writing—original draft, H.S.; conceptualization, methodology, project administration, funding acquisition, writing—review and editing, Y.L. and X.G.; formal analysis, resources, J.L.; validation, supervision, Q.G. All authors have read and agreed to the published version of the manuscript.

**Funding:** This work was financially supported by the National Key R&D Program of China (No. 2020YFD0900201) and the Fundamental Research Funds for the Central Universities of China (No. 842212015).

**Institutional Review Board Statement:** Not applicable.

**Informed Consent Statement:** Not applicable.

**Data Availability Statement:** The data in this study are available from the corresponding author upon reasonable request.

**Acknowledgments:** We are grateful to Qiaohan Wang of Ocean University of China for experimental assistance.

**Conflicts of Interest:** The authors declare no conflict of interest.

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
