# Peer review of "Interactions between Cultivated Gracilariopsis lemaneiformis and Floating Sargassum horneri under Controlled Laboratory Conditions"

_water, doi:10.3390/w14172664_

Round 1
Reviewer 1 Report
The paper consists of a series of experiments that contribute to the explanation of the interactive effects between cultivated G. lemaneiformis and floating S. horneri, a golden-tide-forming seaweed. The results are also expected to provide information to improve mariculture management of G. lemaneiformis, a high-value cultivar and to address marine environmental hazards caused by golden-tide.
The experiments are well designed and conducted and provide enough material for a well-written discussion. In any case, they are more theoretical than applicable and would have to be expanded to provide sufficient information to improve the management of G. lemaneiformis mariculture or to provide sufficient information to prevent environmental hazards caused by golden tide. Nevertheless, the paper contains enough useful and usable information to be accepted, with a note that the English language should be significantly corrected.
Author Response
Response: Thanks a lot for your approval and suggestion for English. Therefore, we have revised the conclusion paragraph of the Discussion in order to highlight the applicable significance in the mariculture management of this macroalga. Also, we carefully revised the manuscript according to the English editing service of MDPI. We hope this version is great enough to be accepted.
Reviewer 2 Report
Dear Authors,
your manuscript is interesting and focused on a stimulating and original topic. The correlation with the growth regulators produced by algae could be an interesting step of the study.
I suggest writing the scientific name of the species in extenso (full Latin binomial) when it is first written in paragraphs or captions.
In my opinion with these few corrections the manuscript could be considered for publication.
Author Response
Response: Your great suggestion is deeply appreciated. We have revised the scientific names of macroalga species when they appeared for the first time in each section.